# Improved Dynamic Compressive and Electro-Thermal Properties of Hybrid Nanocomposite Visa Physical Modification

**DOI:** 10.3390/nano13010052

**Published:** 2022-12-22

**Authors:** Kai Zhang, Xiaojun Tang, Fuzheng Guo, Kangli Xiao, Dexin Zheng, Yunsheng Ma, Qingsong Zhao, Fangxin Wang, Bin Yang

**Affiliations:** 1School of Civil Engineering and Architecture, Suqian University, Suqian 223800, China; 2Beijing Spacecrafts, China Academy of Space Technology, Beijing 100094, China; 3College of Architectural Science and Engineering, Yangzhou University, Yangzhou 225127, China; 4Shandong Chambroad Holding Group Co., Ltd, Binzhou 256500, China; 5School of Aerospace Engineering and Applied Mechanics, Tongji University, Shanghai 200092, China

**Keywords:** physical modification, steric repelling force, mechanical reinforcement, conductive functionality

## Abstract

The current work studied the physical modification effects of non-covalent surfactant on the carbon-particle-filled nanocomposite. The selected surfactant named Triton™ X-100 was able to introduce the steric repelling force between the epoxy matrix and carbon fillers with the help of beneficial functional groups, improving their dispersibility and while maintaining the intrinsic conductivity of carbon particles. Subsequent results further demonstrated that the physically modified carbon nanotubes, together with graphene nanoplates, constructed an effective particulate network within the epoxy matrix, which simultaneously provided mechanical reinforcement and conductive improvement to the hybrid nanocomposite system. For example, the hybrid nanocomposite showed maximum enhancements of ~75.1% and ~82.5% for the quasi-static mode-I critical-stress-intensity factor and dynamic compressive strength, respectively, as compared to the neat epoxy counterpart. Additionally, the fine dispersion of modified fillers as a double-edged sword adversely influenced the electrical conductivity of the hybrid nanocomposite because of the decreased contact probability among particles. Even so, by adjusting the modified filler ratio, the conductivity of the hybrid nanocomposite went up to the maximum level of ~10^−1^–10^0^ S/cm, endowing itself with excellent electro-thermal behavior.

## 1. Introduction

Polymer-matrix nanocomposite materials filled with carbon particles are widely used in aerospace, national defense, transportation, and other industrial fields due to their excellent comprehensive properties, especially in strong structural and functional designability [1,2]. However, the widespread application of nanocomposite materials requires them to be able to satisfy complex and harsh work environments, including various loading and natural conditions. For example, marine nanocomposites must adapt to high-humidity and high-salt environments in boundless seas, and particularly the scientific expedition vessel in polar regions is required to withstand low-temperature ice accretion and drift ice impact beneath the waves [3,4]. On the other hand, the aero nanocomposite will be subject to the impact load and meet the usage requirements under low-temperature icing conditions when an aircraft cruises across the cumulonimbus clouds [5]. Based on the above discussion, the main objective of this work is to solve the low-temperature ice accretion of carbon particle-filled nanocomposite surfaces with the help of their functionality, as well as to improve the mechanical properties so as to realize the overall designability of structure and function.

Nano-scale carbon fillers, represented by carbon nanotubes and graphene, have attracted keen interest in terms of polymer matrix nanocomposites [6,7,8]. These particulate-filled composites display a certain degree of performance improvement, such as in electrical, thermal, electromagnetic, and long-term durability, while relatively less in dynamic mechanical reinforcement [9,10,11,12]. With respect to most high-viscosity matrices filled with nano-scale particles, the outstanding dispersity of carbon fillers in one matrix developed into a huge challenge, which greatly affected the product performance. Generally speaking, nano-scale carbon fillers easily and severely aggregate because of the Van der Waals forces between each other [13]. Furthermore, inorganic carbon particles exhibit poor interfacial interactions with the organic matrix [14]. 

There are two main dispersion strategies: (i) mechanical dispersion [15] and (ii) surface modification assisted dispersion [16,17]. Mechanical dispersion generally consists of ultrasonication, 3-roll grinder, vibrating ball miller, or their combined methods in series or parallel, the advantages of which are simple crafting, low cost, and high recovery rate. The above mechanical methods are accomplished to disperse fillers from each other by shear force or vibration energy. However, this process of mechanical dispersion tends to damage and break the graphite structure, which reduces the specific area of graphene and shortens the length-diameter ratio of carbon nanotube [18]. As we all know, the construction of a conductive network in polymer matrix is strongly dependent on the unidirectional size of carbon fillers, which is considered to be a major disadvantage for function implementation. On the other hand, surface modification assisted strategies can be further divided into chemical and physical methods [19]. In particular, the chemical modification has been widely investigated over the past few decades, leading to the improved dispersibility of carbon fillers. However, two major defects can be summarized for chemical modification methods: (i) most of these are radical, such as the oxidizing reaction with concentrated acids or strong alkalis, and deteriorate the structural characteristics of carbon fillers similar to the mechanical strategy [20,21]; (ii) although several more moderate reactions have been discovered, represented by UV/ozone treatment, a finite number of active sites on the filler surface results in inefficient functionalization [22,23,24]. Accordingly, mechanical reinforcement and functional improvement struggled to be achieved simultaneously for the above chemical modification methods.

In recent years, non-covalent physical modifications based on the surfactant have sprung up like mushrooms [25,26,27]. Breitwieser et al. [26] introduced a new facile protocol for the non-covalent functionalized carbon nanotubes and demonstrated their improved dispersibility in an aqueous buffer. Kulkarni et al [27], in their critical review, stated that the non-covalent physical modifications neither damaged the graphite structure of carbon fillers, nor broke the intrinsic π-bonds of the graphite structure. As a result, the excellent electrical properties of carbon fillers were preserved to achieve the multi-functional features of the as-prepared nanocomposite. At the same time, the physical adsorption of the surfactant on the graphite structure weakened the interfacial tension of the carbon fillers, efficiently stopping the particulate aggregation. Moreover, the physical modification could conquer the Van der Waals’ forces via steric repulsive or electrostatic forces. The uniform dispersity of carbon fillers within the polymer matrix contributed to the overall mechanical reinforcement of the as-prepared nanocomposite. In other words, physical modification could endow both functional improvement and mechanical reinforcement to nanocomposite system. Meanwhile, to increase the carrying efficiency, hybrid carbon fillers with different geometric features have aroused people’s attention. The threadlike carbon nanotube contains a 1-D cannular structure, and graphene with a 2-D flake structure [28,29,30]. For example, Pokharel et al. [30] investigated the conductive and mechanical properties of carbon-nanofiller-filled polyurethane from the points of the particle size, aspect ratio, and filler morphology. They found that an addition of 2 wt% hybrid fillers in polyurethane, at a 1:1:2 ratio between graphene nanoplatelets, carbon black and carbon nanotubes showed the lowest electrical resistivity of ∼106.8 Ω/sq, along with the highest improved mechanical properties. The coordination between different fillers was able to constitute a loosely packed 3-D particulate network in the polymer matrix to raise the reinforcement efficiency. In addition, among various polymers, epoxy resin, one of the most considerable thermosetting matrixes in industrial production, has been selected in quantity as structural components because of its cured cross-link network with high modulus and strength, competitive cost, and outstanding machinability. However, a stiff cross-link structure within the cured epoxy matrix would make it brittle, limiting its applications as a structural component under dynamic loading.

In this work, we selected a kind of non-ionic surfactant named Triton™ X-100 to treat the carbon nanotube surface, which could physically absorb onto the graphite structure of fillers and introduce steric repelling forces between different phases, accordingly improving the dispersion effects of fillers within the epoxy matrix. The modified 1-D carbon nanotube mixed with 2-D graphene nanoplate was then used to construct a 3-D particulate network in the matrix, which aimed to endow both mechanical reinforcement and functional improvement simultaneously to the as-prepared nanocomposite. Subsequently, the physical modification event of Triton™ X-100 was understood through XPS and FT-IR analyses. Meanwhile, the quasi-static fracture properties, dynamic compression behaviors, and conductive functionalities were analysed to demonstrate the overall designability of the structure and function of the as-prepared nanocomposite.

## 2. Experimental and Characterization Details

### 2.1. Materials

Diglycidyl ether of bisphenol-A epoxy resin (EP, YD-128, Kukdo Chemical Co., Ltd., Kunshan, China) with its hardener (Amine type, 5010B, Kukdo Chemical Co., Ltd., Kunshan, China) was adopted to construct the nanocomposite matrix. Meanwhile, multi-layered graphene nanoplates (GNPs, G196540, Aladdin Inc., Los Angeles, CA, USA) and multi-walled carbon nanotubes (CNTs, 698849, Sigma Inc., Berlin, Germany) were purchased as nanofillers to reinforce the matrix. Herein, GNPs have nano-scale thickness and micro-scale plane diameter as a type of 2-D nanoparticle, while CNTs, as 1-D nanoparticles, possess nano-scale diameter and micro-scale length with a huge aspect ratio of hundreds. The detailed parameters of the above materials are listed in Table 1.

### 2.2. Physical Modification

Considering that 1-D CNTs easily and severely agglomerated between each other due to the Van der Waals interaction, a kind of nonionic surfactant (Triton™ X-100, Sigma Inc., Berlin, Germany) was selected to modify the CNTs surface, the critical micelle concentration of which was 0.2–0.9 mM at room temperature [31]. Accordingly, the fortified concentration of surfactant was determined to be ~0.125 mg/mL for the surfactant weight to the volume of acetone solvent (purity > 98%). Subsequently, a certain amount of CNTs was mixed into the diluted surfactant and dispersed ultrasonically at 30 W for 8 h to achieve a uniform suspension of CNT@X, which is the abbreviation for the modified CNTs. Figure 1a accomplished the suspension stability tests for GNP (0.5 wt%) and CNT@X+GNP (3/7 in total 0.5 wt%) nanoparticles in acetone solvent to understand the role of Triton™ X-100. After 6 h at room temperature, GNPs without surfactant treatment displayed a significant sedimentation due to the restacking of lamellar GNPs. However, CNT@X along with GNP particles demonstrated better suspension stability with a smaller supernatant volume. As symbolically depicted in Figure 1b of a single Triton™ X-100 molecule and its structural formula, the hydrophobic octyl group at the Triton™ X-100 molecule was able to interact with CNTs through physical adsorption, while the hydrophilic segment interacted with the epoxy matrix through hydrogen bonding [32,33]. In other words, CNT@X could introduce steric repelling forces to construct a framework between epoxy and GNPs, which contributed to improving the dispersion effects of nanoparticles within the epoxy matrix.

### 2.3. Specimen Fabrication

Two categories of nanocomposite systems filled with CNT@X and/or GNPs were prepared through a convenient procedure: ultrasonic/shear dispersion and room temperature curing. To distinguish them from the variety of nanoparticles dispersed in the epoxy matrix, these two nanocomposite systems are abbreviated as EP/CNT@X+GNPs_x/y and EP/GNPs_z (z = 0.1, 0.25, 0.50, and 1.0 wt% to ensure the improved effects), respectively. In particular, the symbol EP/CNT@X+GNPs_x/y defines the hybrid nanocomposite system, where x/y denotes the weight ratio of CNT@X/GNPs at 0.5 wt% of total content. The weight ratios of CNT@X to GNPs were varied from 3/7 to 5/5 to 7/3, to understand their synergistic effects on the hybrid nanocomposite system. The specified quantity of CNT@X+GNPs mixture was first poured into the heated epoxy matrix, which was then subjected to ultrasonic dispersion for 6 h. To further ensure the dispersion efficiency, a 3-roll grinder (AD IDS50, Changzhou MQ Co., Ltd., China) was subsequently adopted, which provided a strong shear force to the entire mixture by adjusting the angular velocity and the gap setup (5–10 μm) between adjacent rollers. Lastly, after adding the curing agent (1/3 of epoxy by weight), the EP/CNT@X+GNPs mixture was collected into a silastic mold and hardened at room temperature for 12 h. On the other hand, for the single-phase nanocomposite counterpart of EP/GNPs_x, a similar preparation process was conducted for the mechanical and conductive comparisons.

### 2.4. Characterizations and Measurements

#### 2.4.1. XPS and FTIR Characterization

An X-ray photoemission spectra instrument (XPS, VG scientific ESCSLab 250-XI, Walthamm, MA, USA) was employed to demonstrate the surface chemistry of the nanofiller with and without physical modification. Subsequently, the XPS-peak-differentiation-imitating analysis of C 1s was conducted by the XPS Peak Fit V4.1 software.

The functional groups attached to CNT@X+GNPs particles after physical modification were characterized by a FT-IR instrument (Varian, Inc., Palo Alto, CA, USA), which was in transmittance mode using the spectroscopy grade KBr pellet technique. All spectra were recorded in the wavenumber range from 3700 to 700 cm^−1^ with a resolution of 4 cm^−1^. The data were acquired in transmittance mode using the spectroscopy grade KBr pellet technique.

#### 2.4.2. Microstructural Characterization

To verify the microstructural morphology and reinforcement mechanisms, the fracture surfaces of the failed specimens were checked by a field-emission scanning electron microscope (FE-SEM, Zeiss-Supra55, Stuttgart, Germany) at an accelerating voltage of 5 kV.

#### 2.4.3. Electrical and Elastic Measurements

The volt-ampere curves of a series of specimens were measured with a sourcemeter (Keithley 2700, Cleveland, OH, USA) based on a two-point approach. The electrical conductivities were then evaluated by further considering the specimen geometry. On the other hand, an infrared (IR) camera (Flir T420, Boston, MA, USA) was used to characterize the electro-thermal behavior of the as-prepared nanocomposite, and the data processing software ThermaCAM Researcher Pro V2.10 was used to accomplish the temperature distribution analysis. More details can be found in our previous works [15].

The elastic modulus (*E*) and Poisson’s ratio (*v*) were determined with an ultrasonic pulse-echo instrument (Teclab UMS-100 system, France). Ultrasonic transducers (longitudinal wave (*C*_L_): Teclab #L-5-3, 5 MHz; shear wave (*C*_S_): Teclab #S-5-3, 5 MHz) along with the software UMSoft V5.0 were employed to measure the longitudinal wave (*C*_L_) and shear wave (C_S_) velocities. As a result, the elastic modulus (*E*) and Poisson’s ratio (*v*) can be given [34],
(1)$$E=\frac{\left(3Q^{2}-4\right)\rho C_{s}^{2}}{Q^{2}-1}$$
(2)$$v=\frac{Q^{2}-2}{2\left(Q^{2}-1\right)}$$
where *Q* denotes the specific value of *C*_L_/*C*_S_, and the mass density (*ρ*) of the nanocomposite specimens was determined by a density balance with the Archimedes principle (Etnaln ET-320, Dongguan, China).

#### 2.4.4. Quasi-Static Fracture Measurements

The cured nanocomposites were polished into single edge notched bend (SENB) specimens with nominal dimensions of 106 mm × 20 mm × 10 mm (span: 80 mm) for the three-point end notch bending (3P-ENB). As illustrated in Figure 2a, the rectangular specimen was first pre-cracked into a single edge notch at the midspan using a fret saw. Next, the notch tip was sharpened through grinding a thin blade to receive a steady crack initiation site in the subsequent fracture process. 

Subsequently, to extract the mode-I critical-stress-intensity factor (*K*_Ic_) on behalf of quasi-static fracture toughness, the 3P-ENB test was performed for the SENB specimen according to ASTM D5045 [35]. The SENB specimen was loaded into a universal testing machine (Instron 8501, Boston, MA, USA) with a crosshead velocity of 0.5 mm/min and a load cell capacity of 10 kN, as shown in Figure 2b. Significantly, at least four parallel specimens of each nanocomposite category participated in the 3P-ENB test to ensure scientificity. The load-deflection curve was exported upon the specimen until it failed completely, and *K*_Ic_ was then calculated in terms of the fracture load and the geometric dimension:(3)$$K_{\mathrm{IC}}=\frac{SP}{WH^{3/2}}\cdot f(\xi )$$

Herein,
(4)$$f(\xi )=\frac{3\xi^{1/2}\left(1.99-\xi \left(1-\xi \right)\left(2.15-3.93\xi +2.7\xi^{2}\right)\right)}{2\left(1+2\xi \right){\left(1-\xi \right)}^{3/2}}$$
and
(5)$$\xi =\frac{a}{H}$$
where *S*, *P*, *W*, *H*, and *a* are the span, fracture initiation load, width, height, and the pre-crack length of the specimen (Figure 2a), respectively.

The critical strain energy release rate in a plane stain state, *G*_IC_, can be obtained from the following:(6)$$G_{\mathrm{IC}}=\frac{(1-\nu^{2})K_{\mathrm{IC}}^{2}}{E}$$
where *v*, and *E* are Poisson’s ratio and Young’s modulus of nanocomposite specimen.

#### 2.4.5. Dynamic Compressive Measurements

A split Hopkinson pressure bar (SHPB) equipment was employed to study the dynamic compressive response of different nanocomposite categories. As illustrated in Figure 3, the SHPB equipment was mainly composed of a gas gun, striker bar, incident bar, and transmission bar. Herein, the striker bar was inserted into a gas gun barrel and could be accelerated instantaneously towards the incident bar. The striker possessed the same cross-section diameter (16.5 mm) and material attribute (35CrMnSi Alloy, *E* = ~95 GPa) as the incident and transmission bars, which contributed to eliminating the impedance mismatch between adjacent bars, and ensured that the ultimate strength of the bar texture was much higher than the maximum strength of the nanocomposite specimen. In order to accurately capture the stress wave signals across the incident and transmission bars, there was an electric resistance strain gauge attached to the incident bar, while a semiconductor strain gauge with higher sensitivity was attached to the transmission bar, and a red copper slice was selected as the pulse shaping material to achieve a constant strain rate during the load process. Meanwhile, the incident and transmission bars were twice as long as the striker bar to eliminate the superposition of incident and reflected waves. In this work, the incident and transmission bars were determined to be 1500 mm. On the other hand, the nanocomposite specimen was polished into a cylindrical geometry with a cross-section diameter of 13.6 mm (~80% of bar diameter) and longitudinal height of 6.8 mm (0.4–0.6 to specimen diameter), which could decrease the experimental error stemming from the interfacial friction effect [36]. On the other hand, the data acquisition unit included an ultra-dynamic strain indicator (WS-3811/U16, Beijing Wavespectrum Science and Technology Co., Ltd., Beijing, China), an oscilloscope (MDO3104, Tektronix Inc., Suzhou, China), and computing terminal. As the striker with a certain velocity touched the incident bar, an elastic wave was generated and propagated through the nanocomposite specimen, which was then compressed instantaneously. Considering the impedance difference between the incident bar and the nanocomposite specimen, the reflected and transmitted waves would be generated at their contact area. During the dynamic compressive process, the reflected wave returned to the incident bar, while the transmitted wave propagated through the transmission bar. Finally, the waveform was recorded by the data acquisition unit for the following stress–strain analysis.

According to the one-dimension stress wave theory, the stress (*σ*), strain (*ε*), and strain rate (*κ*) of the dynamic compressive test were calculated by [37]: (7)$$\sigma =\frac{A}{2A_{s}}E\left(\varepsilon_{i}+\varepsilon_{r}+\varepsilon_{t}\right)$$
(8)$$\varepsilon =\frac{C}{L_{s}}\int_{0}^{t}\left(\varepsilon_{i}-\varepsilon_{r}-\varepsilon_{t}\right)dt$$
(9)$$\kappa =\frac{C}{L_{s}}\left(\varepsilon_{i}-\varepsilon_{r}-\varepsilon_{t}\right)$$
where *ε_i_*, *ε_r_*, and *ε_t_* are the incident, reflected, and transmitted waves, respectively; *A* and *A_s_* are the cross-section areas of the incident bar and nanocomposite specimen; *L_s_* is the height of a nanocomposite specimen; and *C* is the velocity of the elastic wave across the incident bar.

Based on the stress homogeneity assumption within the nanocomposite specimen, the forces loaded on both ends of the nanocomposite specimen should be equal:(10)$$\varepsilon_{t}=\varepsilon_{i}+\varepsilon_{r}$$

As a result,
(11)$$\sigma =\frac{AE}{A_{s}}\varepsilon_{t}$$
(12)$$\varepsilon =\frac{-2C}{L_{s}}\int_{0}^{t}\varepsilon_{r}dt$$
(13)$$\kappa =\frac{-2C}{L_{s}}\varepsilon_{r}$$

## 3. Results and Discussion

### 3.1. Modification Analysis

XPS analysis was then accomplished to demonstrate the modification mechanism of Triton™ X-100, which could provide more details about the surface structure of carbon particles and identify the functional groups according to the chemical shift observations. An overall scan was first achieved to evaluate the element composition for GNPS and its modified hybrid CNT@X+GNPs, as depicted in Figure 4a,c. Two obvious peaks of C 1s and O 1s were captured in the current scan window (0–800 eV) for both carbon fillers. Meanwhile, Table 2 summarizes the atomic concentration of elements of different carbon fillers. It can be observed that the O 1s of CNT@X+GNPs increased significantly with respect to its counterpart GNPs. According to adsorption theory, we can conclude that the abundance in oxygen content of CNT@X+GNPs was primarily derived from additional nonionic surfactant, indicating that Triton™ X-100 molecules succeeded in absorbing into the nanofiller surface. Furthermore, high-resolution spectra of C 1s were analysed by the software, XPS Peak Fit V4.1, which provided more evidence for the modified mechanism of Triton™ X-100 (Figure 4b,d). The core-level spectrum of C 1s was deconvolved into three peaks at 284.1, 285.2, and 286.3 eV, relating to the C=C, C-C, and C-O structures of GNPs, respectively [38]. The highest cyan peak of C=C was assigned to the sp^2^ structure of graphitic crystal, and the cyan C-C to the sp^3^ hybridization of diamond-like carbon element [39]. In the case of CNT@X+GNPs, there was a concomitant expansion for sp^3^ and C-O structures. The relative percentages of the carbon assignments from **C** 1s analysis are tabulated in Table 3. The dominant bond of pristine CNT itself is the sp^2^-hybridized carbon structure, while the Triton™ X-100 molecule consists of single C-C bonds that are sp^3^-hybridized. Thus, the increased intensity of carbon sp^3^ hybridization signified the physical adsorption of Triton™ X-100 micelles onto the CNT@X surface.

Moreover, the FT-IR spectra was obtained to confirm the above results, as shown in Figure 5. After physical modification for the CNT@X+GNPs particles, several peaks emerged at wavenumbers of 1107–1355 cm^−1^, 1470–1525 cm^−1^, and 2870–3000 cm^−1^, related to the C-O stretching vibration, ring C=C stretching vibration, and aromatic C-H, and -CH_3_, -CH_2_ stretching vibration, respectively. These groups played a significant role in the dispersion process of particles within the epoxy matrix, which introduced the repulsive interactions between modified particles and epoxy matrix. However, there were no clear peaks present on the pristine GNPs surface, which acted as one of carbon allotropes.

### 3.2. Quasi-Static Fracture

In this section, the quasi-static fracture behaviors of the EP/GNPs and EP/CNT@X+GNPs nanocomposite specimens under the 3P-ENB loading condition were investigated, and their failure mechanisms were also interpreted from the fractographic analysis after the specimen failed. As a result, Figure 6 plots the load-deflection curves of a series of nanocomposite specimens during the overall fracture process, and the extracted specific data are summarized in Table 4 quantificationally. The failure characteristics, such as the maximum load and deflection, of various EP/CNT@X+GNPs specimens were all reinforced, obviously due to the existence of hybrid fillers (Figure 6b), indicating a higher load-bearing capacity than their single-phase filled counterparts (Figure 6a). To quantitatively investigate the effects of hybrid/single-phase fillers on the quasi-static fracture properties, the mode-I critical-stress-intensity factor (*K*_Ic_) and fracture energy density (*G*_Ic_) for two categories of nanocomposite materials were calculated according to Equations (3) and (6), as shown in Figure 7. Figure 7a,c exhibit that, although the graphene nanoplate played a positive role in terms of mode-I *K*_Ic_, the reinforced results were limited considering the higher cost of graphene particles. Herein, the EP/GNPs _0.5 case achieved the most significant reinforcement in mode-I *K*_Ic_ among all EP/GNPs specimens (Figure 7a); however, this was only a weak amplification of ~16.6% with respect to the neat matrix. On the other hand, in the EP/GNPs _1.0 case, the slope of the load-deflection curve became steeper in order to reach a maximum failure load (Figure 6a), which contributed to the *G*_Ic_ value (Figure 7b). However, its failure deflection degraded to ~0.3 mm, lower than that of the neat matrix. As d modified CNT@X in varying proportions was added into the nanocomposite system, the fracture toughness of the hybrid EP/CNT@X+GNPs specimens was further enhanced, as shown in Figure 7b,d. The mode-I *K*_Ic_ and *G*_Ic_ of the EP/CNT@X+GNPs_5/5 specimen were determined to be 3.12 ± 0.12 MPa·m^1/2^ and 3.19 ± 0.04 KJ·m^−2^, which exhibited, respectively, the amplification of ~75.1% and ~163.6% when compared to the neat matrix, also full beyond the EP/GNPs category. It was noted that, if the mass ratio between CNT@X and GNPs was adjusted from 7/3 to 3/7, the fracture characteristics of the EP/CNT@X+GNPs_3/7 specimen degraded conveniently, representing a certain synergistic effect. 

To identify the microstructural features and toughness mechanisms of hybrid/single-phase particles within the neat matrix, the fracture surfaces were examined with a FE-SEM instrument, as shown in Figure 8. For the EP/GNPs_1.0 specimen (Figure 8a), the overall fractographic surface was flat and smooth, indicating a weak crack growth resistance [40]. Meanwhile, there were several graphene aggregations to be captured across the observation window. The slender lacerated tails suggested that the graphene aggregations as a crack generator not brake were inefficient to stop the crack growth within the nanocomposite, resulting in the decreased deflection (Figure 6a). On the other hand, the fracture surface of the hybrid nanocomposite exhibited more complex features (Figure 8b–f), and lots of different heights and levels of fragmented fracture surfaces were generated, representing higher crack growth resistance and more fracture energy consumption. In particular, no obvious aggregations were found within the EP/CNT@X+GNPs_7/3 specimen. The fine dispersion of carbon fillers allowed for the uniform deformation for the epoxy matrix and raised the reinforcement efficiency. The FE-SEM images at higher magnification revealed more details about the synergistic effect between CNT@X and GNPs. Figure 8c,d show that several CNT@X particles were attached to the GNPs surface. In other words, the modified CNT@X with excellent dispersity had a chance to be inserted into inter-layers of GNPs and prevent the stacked phenomena, synergistically improving the dispersibility of CNT@X and GNPs. Accordingly, loosely-packed 3-D particulate networks were formed in the neat epoxy to effectively reinforce its fracture toughness, as shown in Figure 9. As mentioned before, if the mass ratio between CNT@X and GNPs was adjusted from 7/3 to 3/7, the fracture characteristics of the EP/CNT@X+GNPs_3/7 specimen degraded conveniently. Figure 8e,f answer this rollback and exhibited several obvious aggregations across the overall fracture surface. It is believed that, within the EP/CNT@X+GNPs_3/7 specimen, the quantity of CNT@X was not enough to interact with more GNPs to construct an integrated CNT@X+GNPs system. The additional GNPs would reaggregate to harm the fracture properties of the EP/CNT@X+GNPs_3/7 specimen. This is also the intrinsic reason why the EP/CNT@X+GNPs_3/7 specimen possessed the most outstanding performance among all nanocomposite specimens.

### 3.3. Dynamic Compression

#### 3.3.1. Dynamic Stress–Strain Relationship under a Strain Rate of 500/s

The dynamic compressive behaviors of different nanocomposite specimens were then investigated with the help of a SHPB equipment in the following text. Herein, two strain rates, 500/s and 1000/s, were considered to study the systematic effects of filler type and content on the as-prepared nanocomposite. Figure 10 first plots the compressive stress–strain curves of different EP/GNPs and EP/CNT@X+GNPs nanocomposite specimens under a strain rate of 500/s. The compressive strength of every EP/GNPs specimen was reinforced to varying degrees due to the existence of GNPs particles, as shown in Figure 10a. Unfortunately, most of these reinforcement events were replaced by the sacrificial failure strain. For example, the peak strength of the stress–strain curve from the EP/GNPs_1.0 case was calculated to be ~126.7 MPa, ~33.40% higher than that of the neat epoxy (Figure 11a), while its failure strain was ~0.035%, ~-39.65% lower than neat epoxy matrix (Figure 11b), indicating that the GNPs contributed more to enhancing the strength, not to toughness. The compressive strength and the matched failure strain of the EP/GNPs_0.25 specimen seemed to achieve an optimal balance among all EP/GNPs specimens, the strength and failure strain of which were ~121.58 MPa and ~0.08%, respectively, and all were superior to the neat epoxy matrix. With the addition of CNT@X into the nanocomposite system, the dynamic compressive properties of the EP/CNT@X+GNPs category increased obviously, including the compressive strength and failure strain (Figure 10b). With respect to different mass ratios between CNT@X and GNPs, the hybrid fillers also exhibited different synergistic toughness to the EP/CNT@X+GNPs category. When the mass ratio was set to 7:3 in the total content of 0.5 wt%, the EP/CNT@X+GNPs_7/3 specimen possessed an optimal reinforcement effect, the peak stress of which reached ~172.9 MPa, and its related failure strain was ~0.08%. However, lower amount of CNT@X was not enough to interact with more GNPs to construct an integrated CNT@X+GNPs system within the EP/CNT@X+GNPs_3/7 specimen. Thus, their compressive strength and failure strain were not obviously increased when compared to the other two hybrid nanocomposite specimens. 

#### 3.3.2. Dynamic Stress–Strain Relationship under a Strain Rate of 1000/s

The dynamic compressive behaviors of different EP/GNPs and EP/CNT@X+GNPs nanocomposite specimens were further investigated under a higher strain rate of 1000/s. As a result, a series of compressive stress–strain curves were also extracted and plotted in Figure 12. When compared to the stress–strain curves under a strain rate of 500/s (Figure 10), the linear stage of stress–strain curves under a higher strain rate became more obvious and adjacent; meanwhile, the nonlinear stage decreased along with the higher failure strength. For the case of EP/GNPs category under a current strain rate of 1000/s (Figure 12a), the dynamic compressive strength and failure strain were close to their neat epoxy matrix, indicating a poor strain rate hardening effect regardless of the GNP contents. Whereas there was a favourable strain rate hardening effect for the hybrid CNT@X+GNPs nanocomposite, as shown in Figure 12b. Noting that, different from the dynamic compressive behaviors under a strain rate of 500/s, although the compressive strength increased obviously, the failure strain decreased simultaneously for the hybrid CNT@X+GNPs nanocomposite (Figure 13). The above discussion demonstrated that the 3-D CNT@X+GNPs filler constructed by 1-D CNT@X and 2-D GNPs within the epoxy matrix was suitably subject to the dynamic loading with a favourable strain rate effect.

### 3.4. Conductive Functionalities

Not limited to mechanical reinforcements, conductive fillers dispersed into insulative polymers also have a chance to form an electrical transport network (inset of Figure 14), which turns the as-prepared nanocomposite into a conductor/semiconductor. Furthermore, as an external voltage is supplied to the conductive nanocomposite, the electro-thermal effect will be activated according to Joule’s law, and many functional applications are expected to be realized for such conductive elements. For example, when an air vehicle passes through a cumulonimbus cloud and encounters large-scale supercooled droplets, the ice accretion will gather on its critical aerodynamic surfaces with a high probability, such as aerofoil. It is worth noting that the Boeing 787 jet equips an original electro-thermal coating for aerofoil deicing based on a metal deposition technology. Particularly, with the increasing proportion of nanocomposites in the aviation industry (Boeing 787 and Airbus 350 already overtopping 50 wt%), conductive nanocomposites are expected to be a competitive candidate for aerofoil deicing [41].

Consequently, Figure 14 characterizes the electrical properties of different nanocomposite categories as a function of filler content. With increasing GNPs content, the electrical conductivities of the EP/GNPs specimens improved significantly, increasing sharply from ~10^−16^–10^−15^ S/cm for the insulative matrix to ~10^−4^–10^−3^ S/cm for the EP/GNPs_1.0 specimen, ~12 orders of magnitude higher than neat epoxy. For the hybrid nanocomposite category, although the total content of CNT@X+GNPs filler was only 0.5 wt%, their electrical conductivities were obviously higher than the EP/GNPs_0.5 specimen, even at an identical filler content. For example, the electrical conductivity of EP/GNPs_0.5 was ~10^−8^–10^−7^ S/cm, and the EP/CNT@X+GNPs_3/7 was ~10^−5^–10^−4^ S/cm, which was already the lowest value among the hybrid specimens, indicating an effective construction for conductive network by CNT@X+GNPs. Meanwhile, by adjusting the weight ratio between GNPs and CNT@X, the synergetic conductivity was then investigated. Different from the dynamic compressive behavior, the electrical conductivity did not improve consistently with increasing the weight ratio of CNT@X, while the EP/CNT@X+GNPs_5/5 achieved a maximum level of ~10^−1^–10^0^ S/cm, higher than the EP/CNT@X+GNPs_7/3 case. Why did more CNT@X contribute to the mechanical reinforcement, however, not to the electrical improvement? We attribute this diversity to the uniform dispersion of CNT@X within the epoxy matrix. The well-dispersed CNT@X reduced the physical contact probability and went against the construction of the conductive network. Whereas the fine dispersion of CNT@X brought the uniform mechanical reinforcements in the EP/CNT@X+GNPs system. On the other hand, if an external voltage is steadily provided to the conductive nanocomposites, their surface temperature will increase and eventually obtain an equilibrium according to the Joule heating effect. We captured this electro-thermal event of conductive nanocomposites with an infrared thermal camera, as shown in Figure 15. Under the action of an external voltage, both rectangular specimens became more highlighted than their surroundings in an infrared mode. Subsequently, the software, *Flir Reporter*, finished the temperature analysis on specimen surface by extracting the stored data in each pixel of the infrared image. Figure 15a,b compare the surface temperature information between EP/GNPs_1.0 and EP/CNT@X+GNPs_5/5 specimens under an identical voltage of 70 V. The EP/CNT@X+GNPs_5/5 specimen showed better electro-thermal behavior than its counterpart, including a higher maximum temperature and a narrower temperature distribution, profiting from the favourable conductive network constructed by CNT@X+GNPs. The above results suggested that the EP/CNT@X+GNPs specimen was expected to be a fine electro-thermal element for de-icing application in some advanced fields.

## 4. Conclusions

The current work presented a strategy to reinforce/improve simultaneously the mechanical/functional behaviors of the as-prepared EP/CNT@X+GNPs nanocomposite, and the following conclusion can be drawn so far:(1)XPS and FT-IR analyses confirmed that the Triton™ X-100 molecule succussed to adsorb physically onto the CNTs surface. In addition, the suspension stability tests suggested that the steric repelling forces between the fillers and the matrix could be introduced to improve their dispersion effects;(2)Although only a total filler content of 0.5 wt% was added into the hybrid nanocomposite, the hybrid EP/CNT@X+GNPs_5/5 specimen achieved a maximum mode-I *K*_Ic_ of 3.12 ± 0.12 MPa·m^1/2^ and *G*_Ic_ of 3.19 ± 0.90 KJ·m^2^. The fractographic examination revealed rougher fracture surfaces in EP/CNT@X+GNPs nanocomposites than in their counterparts, suggesting greater energy dissipation and resistance during the crack propagation. Noting that, for the EP/CNT@X+GNPs_3/7 specimen, less CNT@X could not be combined with too much graphene to form an integrated CNT@X+GNPs system, causing the degradation of fracture properties;(3)The dynamic compressive properties were improved by increasing the ratio of CNT@X in the hybrid fillers at either 500/s or 1000/s. Herein, the EP/CNT@X+GNPs_7/3 specimen possessed an optimal reinforcement effect, the peak stress of which reached ~172.9 MPa, and its related failure strain was ~0.08% under 500/s;(4)In the case of conductive functionalities, the EP/CNT@X+GNPs_5/5 specimen showed better electro-thermal efficiency than its counterpart under an identical voltage of 70 V, and its surface temperature was over 100 °C in an infrared mode.

## Figures and Tables

**Figure 1 nanomaterials-13-00052-f001:**
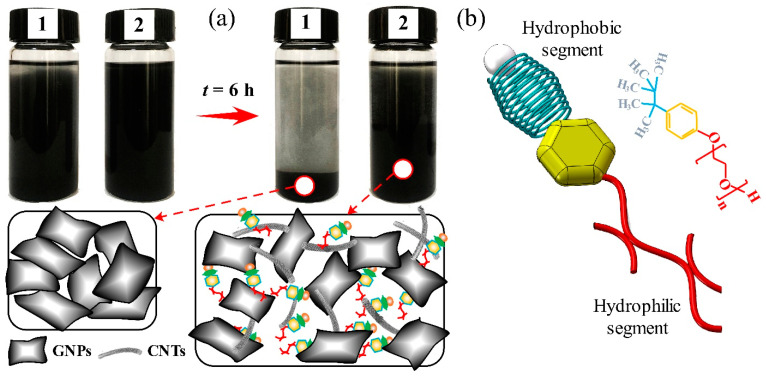
Physical modification: (**a**) suspension stability tests for 1: GNPs and 2: CNT@X+GNPs nanoparticles in acetone solvent, and (**b**) schematic diagram showing a single Triton™ X-100 molecule and its chemical structure.

**Figure 2 nanomaterials-13-00052-f002:**
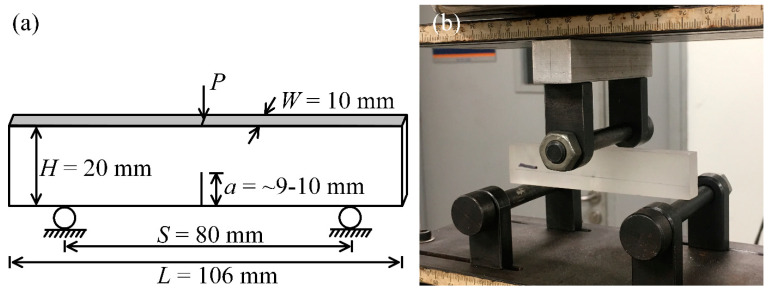
Schematic diagram showing (**a**) the three-point end notch bending (3P-ENB) test for the single edge notched bend (SENB) specimen and (**b**) its testing scene on a universal testing machine.

**Figure 3 nanomaterials-13-00052-f003:**
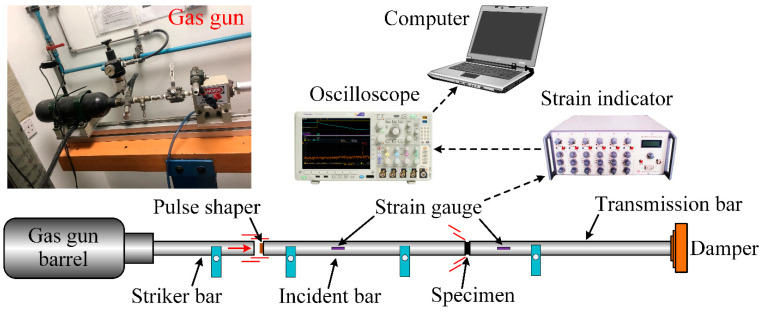
Schematic diagram showing the split Hopkinson pressure bar (SHPB) equipment and its data acquisition unit.

**Figure 4 nanomaterials-13-00052-f004:**
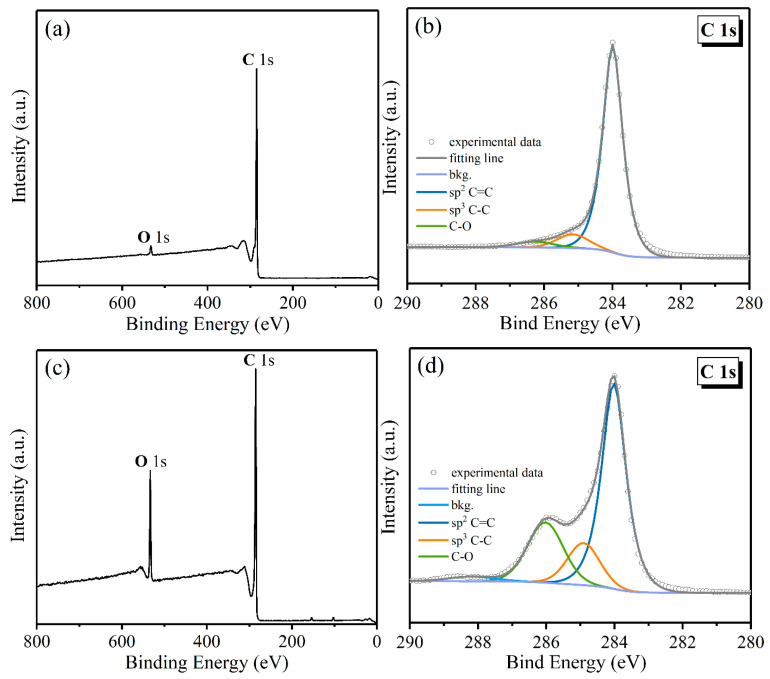
XPS scans for (**a**) GNPs and (**b**) CNT@X+GNPs fillers and (**c**,**d**) their high-resolution scan spectra of C 1s peaks.

**Figure 5 nanomaterials-13-00052-f005:**
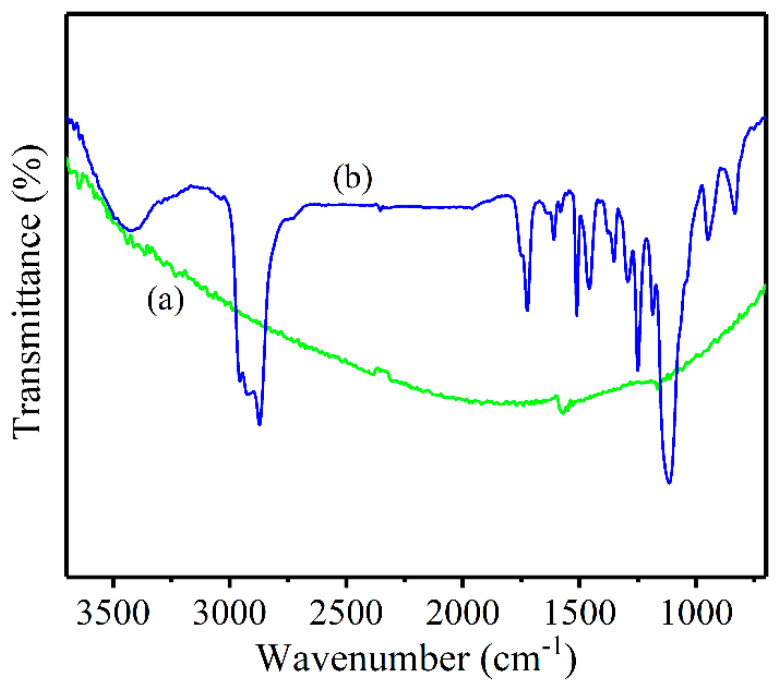
FT-IR spectra of (**a**) GNPs and (**b**) CNT@X+GNPs particles.

**Figure 6 nanomaterials-13-00052-f006:**
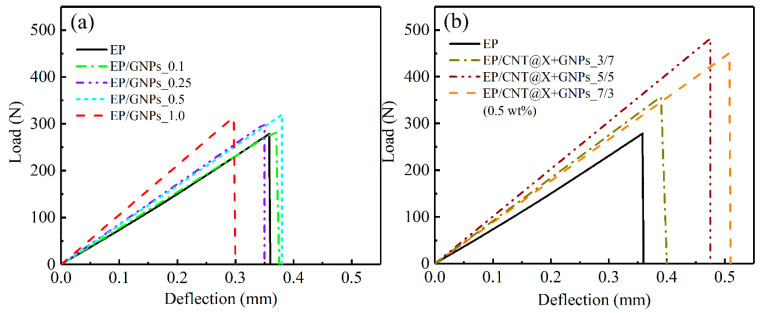
Load-deflection curves of the (**a**) EP/GNPs and (**b**) EP/CNT@X+GNPs nanocomposite specimens under the quasi-static 3P-ENB loading condition.

**Figure 7 nanomaterials-13-00052-f007:**
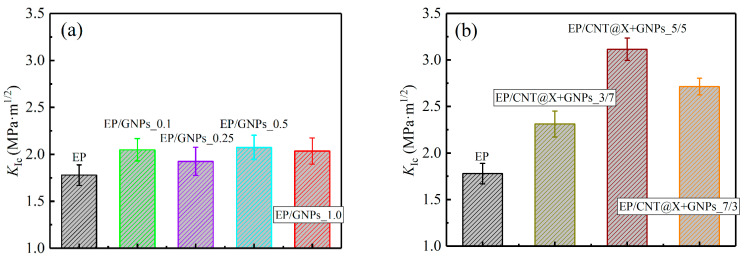
Quasi-static 3P-ENB fracture parameters: (**a**,**b**) mode-I critical-stress-intensity factor (*K*_Ic_) and (**c**,**d**) fracture energy density (*G*_Ic_) of the EP/GNPs and EP/CNT@X+GNPs nanocomposite specimens.

**Figure 8 nanomaterials-13-00052-f008:**
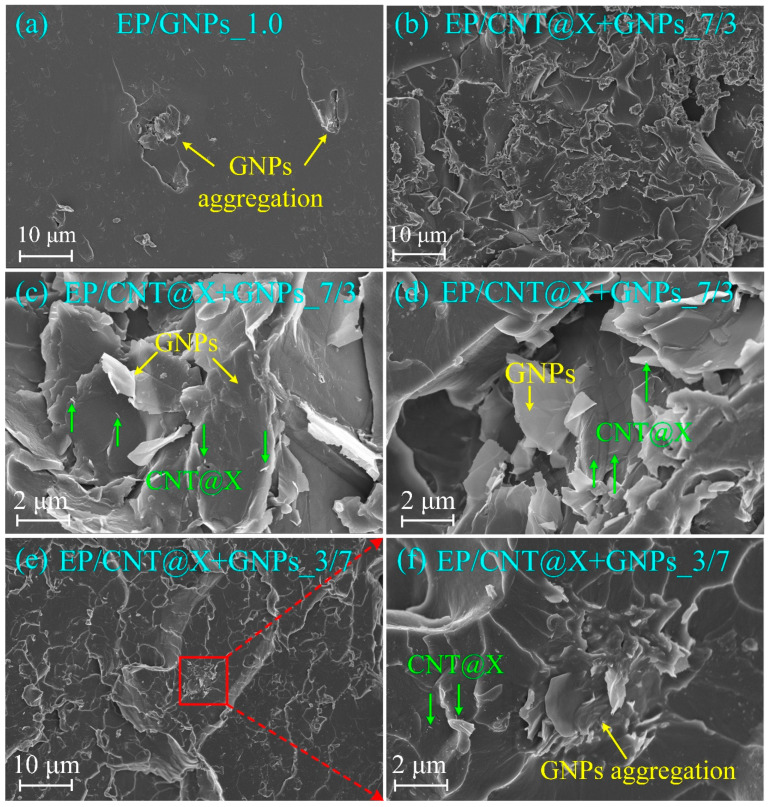
Fracture micrographs for different nanocomposite specimens under the quasi-static 3P-ENB loading condition.

**Figure 9 nanomaterials-13-00052-f009:**
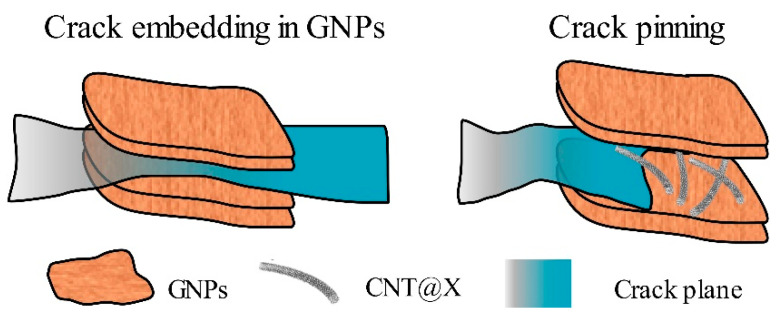
Schematic diagram showing the interaction of the crack front with GNPs and CNT@X+GNPs particles.

**Figure 10 nanomaterials-13-00052-f010:**
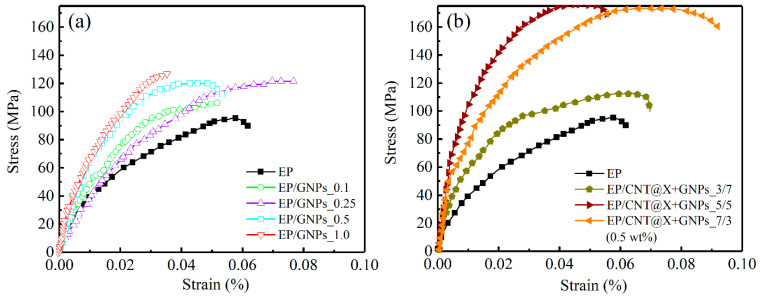
Dynamic compressive stress–strain curves of the (**a**) EP/GNPs and (**b**) EP/CNT@X+GNPs nanocomposite specimens under a strain rate of 500/s.

**Figure 11 nanomaterials-13-00052-f011:**
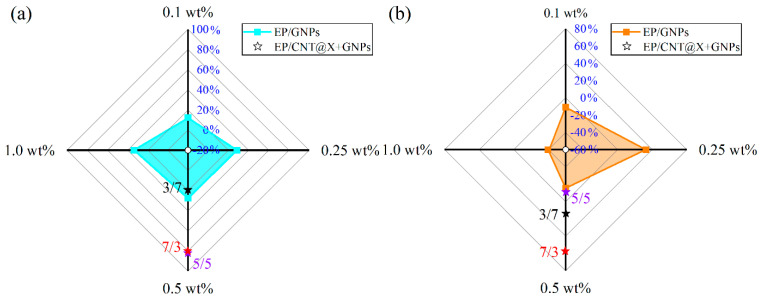
Radar plot for improved (**a**) dynamic compressive strength and (**b**) failure strain with respect to neat epoxy for different nanocomposite systems under a strain rate of 500/s.

**Figure 12 nanomaterials-13-00052-f012:**
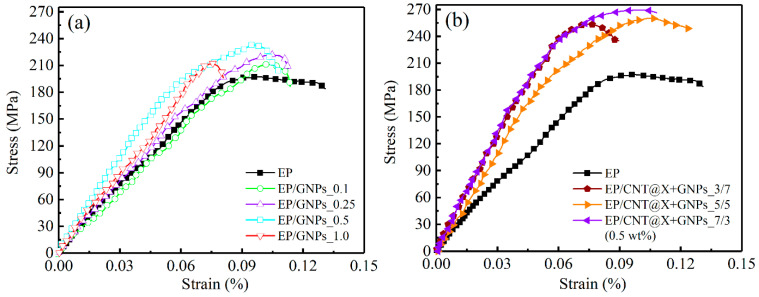
Dynamic compressive stress–strain curves of the (**a**) EP/GNPs and (**b**) EP/CNT@X+GNPs nanocomposite specimens under a strain rate of 1000/s.

**Figure 13 nanomaterials-13-00052-f013:**
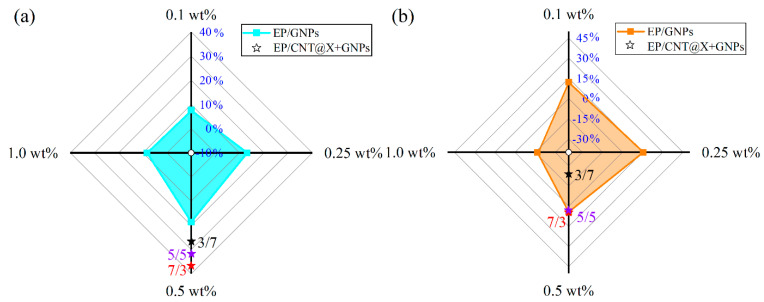
Radar plot for improved (**a**) dynamic compressive strength and (**b**) failure strain with respect to neat epoxy for different nanocomposite systems under a strain rate of 1000/s.

**Figure 14 nanomaterials-13-00052-f014:**
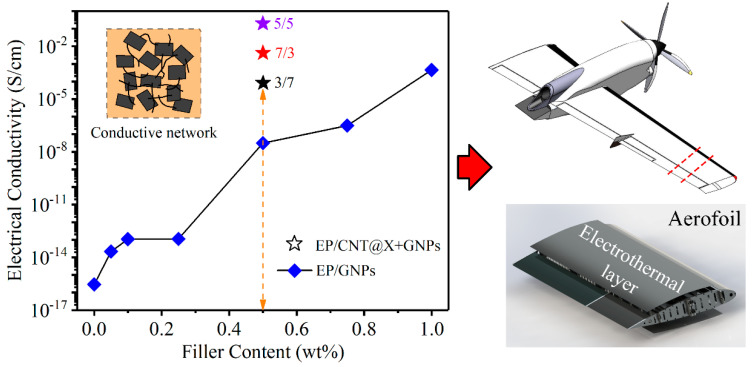
Electrical conductivities of the EP/GNPs and EP/CNT@X+GNPs nanocomposite specimens as a function of filler content, and their expected application in the aviation industry.

**Figure 15 nanomaterials-13-00052-f015:**
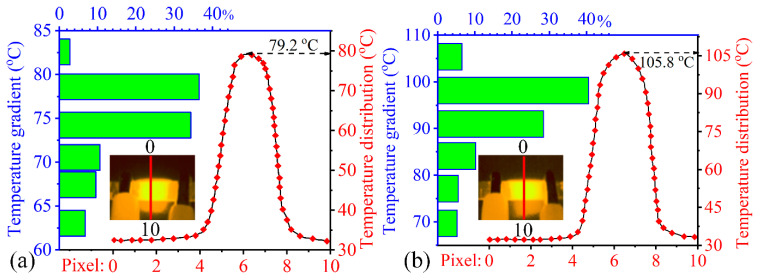
Surface temperature analysis for (**a**) EP/GNPs_1.0 and (**b**) EP/CNT@X+GNPs_5/5 specimens under an applied voltage of 70 V, including the line temperature distribution (dash-dotted line) along the pixel length: 0 → 10 and the statistical distribution (histogram) of surface temperature between two electrodes.

**Table 1 nanomaterials-13-00052-t001:** Physicochemical properties and production technology of raw materials.

Materials	Conductivity	Surface Area	Viscosity	Density	Form	Technology
GNPs	~87 S/cm	~732 m^2^/g	-	~0.02 g/mL	Powder	CVD
CNTs	~128 S/cm	~220 m^2^/g	-	~2.1 g/mL	Powder	CVD
Epoxy	-	-	~7000 Mpa·s (25 °C)	~1.3 g/cm^3^	Liquid	One step synthesis
Hardener	-	-	~11 Mpa·s (25 °C)	~0.96 g/cm^3^	Liquid	-

**Table 2 nanomaterials-13-00052-t002:** Atomic concentration of elements in GNPs and CNT@X+GNPs.

	C 1s	O 1s	Impurities	O/C ratio
GNPs	96.61%	2.87%	0.52%	0.030
CNT@X+GNPs	85.37%	13.46%	1.17%	0.158

**Table 3 nanomaterials-13-00052-t003:** Related percentages of carbon assignments from C 1s analysis.

	Binding Energy (eV) and Assignments		
	sp^2^ C=C	sp^3^ C-C	C-O
GNPs	67.43%	11.69%	20.88%
CNT@X+GNPs	48.27%	23.51%	28.22%

**Table 4 nanomaterials-13-00052-t004:** Quasi-static mechanical properties of the EP/GNPs and EP/CNT@X+GNPs nanocomposite specimens.

Specimens	Maximum Load	Maximum Deflection	Increment Proportion of Load	Increment Proportion of Deflection
EP	279.01 N	0.36 mm	-	-
EP/GNPs_0.1	285.45 N	0.37 mm	2.31%	2.78%
EP/GNPs_0.25	300.69 N	0.35 mm	7.77%	−2.75%
EP/GNPs_0.5	319.84 N	0.38 mm	14.63%	5.56%
EP/GNPs_1.0	314.69 N	0.30 mm	12.79%	−16.67%
EP/CNT@X+GNPs_3/7	357.67 N	0.39 mm	28.19%	8.33%
EP/CNT@X+GNPs_5/5	481.75 N	0.48 mm	72.66%	33.33%
EP/CNT@X+GNPs_7/3	450.80 N	0.51 mm	61.57%	41.67%

## Data Availability

The data presented in this study are available on request from the corresponding author.

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
