# Peer review of "Improved Dynamic Compressive and Electro-Thermal Properties of Hybrid Nanocomposite Visa Physical Modification"

_nanomaterials, 2022, doi:10.3390/nano13010052_

Round 1

Reviewer 1 Report

Reviewed paper “Improved dynamic compressive and electro-thermal properties of hybrid nanocomposite visa physical modification” deals with an interesting topic and can be interesting for readers of Nanomaterials journal. This manuscript seems to provide interesting information about electro-thermal properties of hybrid nanocomposite so it may be interest to the materials community.

The body text contains 13 figures and 2 tables – figures are good quality. English of the paper is rather poor – in my opinion the language of the paper should be a little improved.

I find some mistakes for example:

-      Introduction chapter should be correct. Authors should include new information about topic of a paper. More information based on worldwide (global) study. The list of references should also be changed. Note the following suggestions related to the literature cited.

-      Please describe all equipment and/or software used in the experiment – please insert model of equipment (manufacturer, city, country).

-      More over minimum 24 cited papers out of 38 (over 63 %) are wrote by authors from Asia (most of them from China and some from others countries). Are the topics covered in this article are mostly carried out by research centers from Asia? I ask for a reliable verification of global research in this field. I propose to add some new (from the last 5 years) publications on this topic. Author should include several modern papers (also from Europe and America).

-      In the list of references I found 3 papers of the Authors of reviewed paper. Please indicate the differences in the studies presented in the cited article and their relation to the presented topic and/or are these the first research results published by this team.

-      Figures (captions) – In the case of multi-component figures, the figure caption shall first give the name of the figure followed by references (a), (b) and so on (Figures: 1, 2 and 6).

-      Figure 3 –the fifure should be enlarged for better readability.

-      In the whole paper (text) you write for example 96.61% (for example tables) – you should write this value and symbol with a space as ‘96.61 %’.

-      Please prepare a literature review of a paper according to the guidelines of the Nanomaterials journal.

The paper is potentially interesting but the current version needs MINOR revision to be accepted for publication in Nanomaterials journal.

Author Response

Reviewer #1: Reviewed paper “Improved dynamic compressive and electro-thermal properties of hybrid nanocomposite visa physical modification” deals with an interesting topic and can be interesting for readers of Nanomaterials journal. This manuscript seems to provide interesting information about electro-thermal properties of hybrid nanocomposite so it may be interest to the materials community. The body text contains 13 figures and 2 tables – figures are good quality. English of the paper is rather poor – in my opinion the language of the paper should be a little improved.

The paper is potentially interesting but the current version needs MINOR revision to be accepted for publication in Nanomaterials journal.

1) Introduction chapter should be correct. Authors should include new information about topic of a paper. More information based on worldwide (global) study. The list of references should also be changed. Note the following suggestions related to the literature cited. More over minimum 24 cited papers out of 38 (over 63 %) are wrote by authors from Asia (most of them from China and some from others countries). Are the topics covered in this article are mostly carried out by research centers from Asia? I ask for a reliable verification of global research in this field. I propose to add some new (from the last 5 years) publications on this topic. Author should include several modern papers (also from Europe and America).

√Response: The dynamic mechanical and electro-thermal functional properties of polymer-matrix composites have attracted wide attention from all over the world. According to this comment, we checked the original manuscript and supplemented more references from Europe and America in recent years. Thanks for your constructive suggestion. Please see the new reference list in the revised manuscript.

2) Please describe all equipment and/or software used in the experiment – please insert model of equipment (manufacturer, city, country).

√Response: According to this comment, we supplemented the equipment and software information as detailed as possible, such as 3-roll grinder, XPS, FT-IR, FE-SEM, sourcemeter, infrared camera, ultrasonic pulse-echo instrument, and their data processing software. For the reader's convenience, we supplemented a 2.4. Characterizations and measurements section in the revised manuscript.

3) In the list of references, I found 3 papers of the Authors of reviewed paper. Please indicate the differences in the studies presented in the cited article and their relation to the presented topic and/or are these the first research results published by this team.

√Response: In the original manuscript, we cited two of our own papers, as follows:

[15] Wang, F.; Zhang, K.; Liang, W.; Wang, Z.; Yang, B. Experimental and analytical studies on the flexible, low-voltage electrothermal film based on the multi-walled carbon nanotube/polymer nanocomposite. Nanotechnology 2018, 30, 065704. [CrossRef]

[38] Wang, F.; Tay, T.E.; Sun, Y.; Liang, W.; Yang, B. Low-voltage and-surface energy SWCNT/poly (dimethylsiloxane)(PDMS) nanocomposite film: Surface wettability for passive anti-icing and surface-skin heating for active deicing. Compos. Sci. Technol. 2019, 184, 107872. [CrossRef]

For the Reference 15, we used a 3-roll grinder to disperse the pristine CNTs into poly(m-phenyleneisophthalamide) matrix. Subsequently. The electro-thermal behavior of as-prepared composite was investigated systemically. In the current work, we mentioned two main strategies: (i) mechanical dispersion and (ii) surface modification assisted dispersion in the Introduction section. 3-roll grinder in our own work (Reference 15) as a typical mechanical dispersion method was cited.

On the other hand, in the 3.4. Conductive functionalities section of the original manuscript, we look ahead to the application of conductive nanocomposite in the aerospace field. The Reference 38 just introduced several de-icing methods for aircraft, and investigated the surface wettability for passive anti-icing and surface-skin heating for active deicing.

In the revised manuscript, we have replaced the Reference 38 of the original manuscript with another suitable work some else. Please see the new Reference 41 of the revised manuscript.

[41] Rathod, V.T.; Kumar J.S.; Jain A. Polymer and ceramic nanocomposites for aerospace applications. Appl. Nanosci., 2017, 7, 519. https://doi.org/10.1007/s13204-017-0592-9

4) Figures (captions) – In the case of multi-component figures, the figure caption shall first give the name of the figure followed by references (a), (b) and so on (Figures: 1, 2 and 6).

√Response: In the revised manuscript, we have modified the figure captions to first give the name of the figure followed by (a), (b). Thank you for the reminder!

5) Figure 3 –the figure should be enlarged for better readability.

√Response: In the revised manuscript, we enlarged Figure 2 (i.e., Figure 3 of the original manuscript) and Figure 3 to 16-cm width, and improved the DPI to 1200 × 1200.

6) In the whole paper (text) you write for example 96.61% (for example tables) – you should write this value and symbol with a space as ‘96.61 %’.

√Response: In the revised manuscript, we checked the whole main text carefully, and modified every incorrect description, writing the value and its value with a space.

7) Please prepare a literature review of a paper according to the guidelines of the Nanomaterials journal.

√Response: In the revised manuscript, we have introduced in detail several significant works in the Introduction section, including the following works. Thanks for this constructive comment!

[26] Breitwieser A, Sleytr U B, Pum D. A New Method for Dispersing Pristine Carbon Nanotubes Using Regularly Arranged S-Layer Proteins. Nanomaterials, 2021, 11(5): 1346.

[27] Kulkarni H B, Tambe P, M. Joshi G. Influence of covalent and non-covalent modification of graphene on the mechanical, thermal and electrical properties of epoxy/graphene nanocomposites: a review[J]. Composite Interfaces, 2018, 25(5-7): 381-414.

[30] J Pokharel P, Xiao D, Erogbogbo F, et al. A hierarchical approach for creating electrically conductive network structure in polyurethane nanocomposites using a hybrid of graphene nanoplatelets, carbon black and multi-walled carbon nanotubes[J]. Composites Part B: Engineering, 2019, 161: 169-182.

Thank you very much for careful review. Meanwhile, we tried our best to check the full text carefully and revised the grammar and spelling mistakes. If there is anything we cannot clear up, please let us know.

Reviewer 2 Report

The manuscript under the title: “Improved dynamic compressive and electro-thermal properties of hybrid nanocomposite visa physical modification” is in line with Nanomaterials journal. This topic is relevant and will be of interest to the readers of the journal. It based on original research. This research has scientific novelty and practical significance. The article has a typical organization for research articles.
Before the publication it requires significant improvements, especially:

1.    1. The "Introduction" section: The functionalization of nanodispersed fillers is very effective, and, as you rightly noted, it can be physical and/or chemical. I believe that it is necessary to add specific examples in the "Introduction" section and indicate their advantages and disadvantages. I think the related references should be cited corresponding to each aspect, e.g. (but not limited to these), which will undoubtedly improve the "Introduction" section:

- Russ J Appl Chem 86, 765–771 (2013). https://doi.org/10.1134/S107042721305025X

- Polymer Composites, 36, 1891-1898 (2015). https://doi.org/10.1002/pc.23097

- Polymers 202214(21), 4594; https://doi.org/10.3390/polym14214594

- Appl. Polym. Sci. 2019, 136, 47410, https://doi.org/10.1002/app.47410

- Polymers 202214(21), 4594; https://doi.org/10.3390/polym14214594

- Polymer Composites, 39, E2552-E2561 (2018). https://doi.org/10.1002/pc.24832

  1. Section 2.1. It is necessary to add the physicochemical characteristics of components - give a table with the main physicochemical and technological properties of epoxy resin, hardener, GNPs and CNTs.
  2. Figure 2 and Table 1-2 and their description should be moved to section 3.
  3. Section 2.2. it is necessary to describe the functionalization technique (concentration, solvent, reaction conditions, etc.) in more detail in order for this experiment to be reproducible and understandable.
  4. Section 2.4. for formula (4), it is necessary to indicate how n and E (Poisson’s ratio and Young’s modulus) were determined (calculated).
  5. It would be good to confirm the data in Fig. 2 and Tables 1-2, for example, with the FTIR data.

Author Response

Reviewer #2: The manuscript under the title: “Improved dynamic compressive and electro-thermal properties of hybrid nanocomposite visa physical modification” is in line with Nanomaterials journal. This topic is relevant and will be of interest to the readers of the journal. It based on original research. This research has scientific novelty and practical significance. The article has a typical organization for research articles. Before the publication it requires significant improvements, especially:

1) The "Introduction" section: The functionalization of nanodispersed fillers is very effective, and, as you rightly noted, it can be physical and/or chemical. I believe that it is necessary to add specific examples in the "Introduction" section and indicate their advantages and disadvantages. I think the related references should be cited corresponding to each aspect, e.g. (but not limited to these), which will undoubtedly improve the "Introduction" section:

- Russ J Appl Chem 86, 765–771 (2013). https://doi.org/10.1134/S107042721305025X

- Polymer Composites, 36, 1891-1898 (2015). https://doi.org/10.1002/pc.23097

- Polymers 2022, 14(21), 4594; https://doi.org/10.3390/polym14214594

- Appl. Polym. Sci. 2019, 136, 47410, https://doi.org/10.1002/app.47410

- Polymers 2022, 14(21), 4594; https://doi.org/10.3390/polym14214594 Repeated

- Polymer Composites, 39, E2552-E2561 (2018). https://doi.org/10.1002/pc.24832

√Response: In Russ J Appl Chem 86, 765–771 (2013). https://doi.org/10.1134/S107042721305025X, Burmistrov et al. evaluated the influence of surface modification of the potassium titanate particles on their properties, interaction with epoxy diane oligomers, and strength characteristics of epoxy compounds. It was shown that the effectiveness of the finishing additive depends essentially on the nature of its interaction with the polymeric binder. The finishing agents A-187 and AGM-9 enter into chemical interaction with the components of the epoxy compounds, leading to substantially improved

strength properties of the compounds.

In Polymer Composites, 36, 1891-1898 (2015). https://doi.org/10.1002/pc.23097, Hameed et al. studied the effects of functionalization and weight fraction of mutliwalled carbon nanotubes (CNTs) were investigated on mechanical and thermomechanical properties of CNT/Epoxy composite. They found that Among different types and loading levels of CNTs explored, nanocomposites containing 0.5 wt% aminefunctionalized CNTs offer the best combination of thermal and mechanical properties with slight compromise on strain to fracture.

In Polymers 2022, 14(21), 4594; https://doi.org/10.3390/polym14214594, Shcherbakov et al. The possibility of using microwave radiation at various stages of obtaining an unsaturated polyester composite modified with carbon nanotubes was studied. The improvement of the thermal stability of composites was noted upon the addition of MWCNTs and microwave modification at the stage of the oligomer and composite.

In Appl. Polym. Sci. 2019, 136, 47410, https://doi.org/10.1002/app.47410, aminopropyl trimethoxysilane as an interfacial modifier was introduced on the surface of graphene nanoplatelets. The effects of the silane-modified graphene loading (0, 0.05, 0.1, 0.3, and 0.5 wt %) and silane modification on the tensile, compressive, interlaminar shear stress, and tribological properties of the epoxy-based nanocomposites were investigated by Amirbeygi et al. The outcome of this study suggests that the incorporation of silane-modified graphene as a reinforcing phase is an effective strategy for enhancing the tribological response of epoxy-based specimens.

In Polymer Composites, 39, E2552-E2561 (2018), https://doi.org/10.1002/pc.24832, the MD simulation was combined with experimental methods to study the microstructure-property relationship in MWCNT-NH2 modified two tri-functional epoxy-based systems. The MD simulation indicated that a stronger interfacial strength between MWCNT-NH2 and n, n-diglycidyl-4-glycidy-loxyaniline molecule was formed in contrast to that of epoxy-based sample, no matter with non-covalent or covalent interaction, which was derived to the presence of benzene ring and nitrogen atom in the backbone of n, n-diglycidyl-4-glycidy-loxyaniline molecule.

In the revised manuscript, we cited the above significant works. Please see the References [8, 9, 12, 16, 22].

2) Section 2.1. It is necessary to add the physicochemical characteristics of components - give a table with the main physicochemical and technological properties of epoxy resin, hardener, GNPs and CNTs.

√Response: By looking up the handbook of production scheduling or testing these on our own, we supplemented a new Table 1 in 2.1. Materials section. Herein, we provided some important information about epoxy resin, hardener, GNPs and CNTs, which were related to the topic of the current work.

3) Figure 2 and Table 1-2 and their description should be moved to section 3.

√Response: According to this comment, we moved Figure 2 and Tables 1-2 to section 3. Results and discussion as 3.1. Modification analysis in the revised manuscript. Thanks.

4) Section 2.2. it is necessary to describe the functionalization technique (concentration, solvent, reaction conditions, etc.) in more detail in order for this experiment to be reproducible and understandable.

√Response: In the 2.2. Physical modification section of the revised manuscript, we added the solvent type and its purity, as well as the nanoparticle concentration for the suspension stability tests. The physical modification temperature was also provided. On the other hand, we also supplement a new Reference 31 to introduce the functionalization technique of nonionic surfactant, Triton™ X-100.

[31] Hamze S, Berrada N, Cabaleiro D, et al. Few-layer graphene-based nanofluids with enhanced thermal conductivity[J]. Nanomaterials, 2020, 10(7): 1258. https://doi.org/10.3390/nano10071258

5) Section 2.4. for formula (4), it is necessary to indicate how n and E (Poisson’s ratio and Young’s modulus) were determined (calculated).

√Response: The elastic modulus (E) and Poisson’s ratio (v) were determined with ultrasonic pulse-echo instrument and Archimedes' balance, as shown in Figure R1. In the revised manuscript, we supplemented a new 2.4.3. Electrical and elastic measurements section to describe this point.

Figure R1. Density balance with Archimedes' principle and ultrasonic pulse-echo characterization system

6) It would be good to confirm the data in Fig. 2 and Tables 1-2, for example, with the FTIR data.

√Response: Upon an online meeting, we agreed that this is a constructive comment. Therefore, in the revised manuscript, we supplemented a FT-IR test to confirm the functional groups attached to carbon particle, as shown in Figure 5 of the revised manuscript. The FT-IR results of CNT@X+GNPs showed several wavenumber peaks, such as 1100-1350 cm-1 related to the C-O stretching vibration, which was also found and changed obviously in XPS analysis for CNT@X+GNPs.

Thank you very much for careful reviews, which is a strong help for this and our future works! If you have any other questions, please let us know.

Reviewer 3 Report

The effect of carbon nano particles on the properties of epoxy resin matrix composites is systematically studied. The paper is well designed to analyze the improvement effect of composites with some professional charts and important data. Before accepting the paper, the following comments are suggested to respond to improve the quality of the paper.

1.     The writing of the abstract does not convey some important information on the results found in this paper. The research significance of current work should be further condensed and more qualitative and quantitative results are preferable. In addition, the interaction mechanism between epoxy carbon filler, as well as their enhancement of mechanical properties and functional improvement of composite materials, should be further clarified. According to the above comments, it is suggested to rewrite the abstract.

2.     Introduction, the following comments should be answered further. (1) In the first paragraph, please clarify the type (carbon-fillers) of nanocomposites, which should also be matched with their excellent comprehensive properties, especially in strong structural and functional designability. This is because not all nanomaterials have the above performance and advantages. (2) This paper mainly uses epoxy resin as the matrix. However, the reviewer did not see any analysis and summary on the performance, advantages and application fields of epoxy resin. (3) In addition to the improvement of mechanical properties and thermal properties, the addition of carbon fillers can also improve the long-term durability of composites. It is recommended to add relevant analysis and summary. Please review the latest research work related to relevant comments below for necessary supplement, such as carbon fillers: Nanomaterials, 2021, 11:1234 and Polymers, 2022, 14(10), 1935. Epoxy application: Composite Structures, 2022, 293, 115719.

3.     Is the non-covalent physical modifications reliable to improve the performance of composite? How to verify the long-term reliability for the physical modifications? Generally speaking, the chemical modifications through the formation of chemical bonds may be more stable and reliable.

4.     Please add the basic performance and parameters of raw materials and other relevant information in Part 2.1.

5.     Figure 2 and table 1-2 shows the XPS curves and data before and after the modification. They should belong to the result part and should be put in the result and discussion part. This part focuses on sample preparation and performance test methods.

6.     How to determine the addition content of nanoparticles in Section 2.3? Is there any relevant reference basis?

7.     In addition to the macro performance testing, the micro performance testing and analysis are also critical for revealing the influence of nanoparticles on the properties of composites. It is suggested to add the test methods and details of t scanning electron microscopy.

8.     Figure 5 shows that the load-displacement curves of composites through quasi-static 3P-ENB test. However, the control strength and corresponding improvement proportion should be quantified into a table, so that the reader can understand the relevant information more intuitively.

9.     Fig. 6 shows that the fracture toughness of the composite is improved to different degrees after adding different nanoparticles. Please provide some explanations of the improvement mechanism. It would be better if you could provide the mechanism diagrams.

10.  In part 3.2.1 and 3.2.2, does the strain rate have a significant effect on the stress-strain curve of material? Why do you study the effect of strain rate?

11.  The conclusions should be further cleared with some quantitative results based on the important phenomena and findings in this paper. 3~4 conclusions are suggested.

Author Response

Reviewer #3: The effect of carbon nano particles on the properties of epoxy resin matrix composites is systematically studied. The paper is well designed to analyze the improvement effect of composites with some professional charts and important data. Before accepting the paper, the following comments are suggested to respond to improve the quality of the paper.

1) The writing of the abstract does not convey some important information on the results found in this paper. The research significance of current work should be further condensed and more qualitative and quantitative results are preferable. In addition, the interaction mechanism between epoxy carbon filler, as well as their enhancement of mechanical properties and functional improvement of composite materials, should be further clarified. According to the above comments, it is suggested to rewrite the abstract.

√Response: In the revised manuscript, we rewrote the abstract and described several important mechanisms and finding data for the mechanical properties and functional improvement of the nanocomposite. Thanks for your professional comment!

2) Introduction, the following comments should be answered further. (1) In the first paragraph, please clarify the type (carbon-fillers) of nanocomposites, which should also be matched with their excellent comprehensive properties, especially in strong structural and functional designability. This is because not all nanomaterials have the above performance and advantages. (2) This paper mainly uses epoxy resin as the matrix. However, the reviewer did not see any analysis and summary on the performance, advantages and application fields of epoxy resin. (3) In addition to the improvement of mechanical properties and thermal properties, the addition of carbon fillers can also improve the long-term durability of composites. It is recommended to add relevant analysis and summary. Please review the latest research work related to relevant comments below for necessary supplement, such as carbon fillers: Nanomaterials, 2021, 11:1234 and Polymers, 2022, 14(10), 1935. Epoxy application: Composite Structures, 2022, 293, 115719.

√Response: (1) In the first paragraph of the revised Introduction, we stated the type of nanocomposite, i.e., polymer-matrix nanocomposite materials filled with carbon particles;

(2) In the second paragraph from the bottom of the Introduction, we analysed the performance, advantages, and structural application of the epoxy matrix. We also raised the existing question of the epoxy matrix to reflect the significance of this work. Thanks for your professional comment;

(3) In the Nanomaterials, 2021, 11:1234, the mechanical and water uptake properties of epoxy nanocomposites with surfactant modified MWCNTs were investigated. The homogenous distribution of the CNTs in the epoxy matrix was achieved by wrapping the surfactant onto the CNTs. Thermal, mechanical, morphological, contact angle, and water uptake tests were performed to reveal the improvement mechanism of the CNTs on the epoxy matrix. The incorporation of the CNTs into the epoxy could impart a reduction in the wettability on the surface of the epoxy/CNT nanocomposite, leading to an increase in the contact angle and a reduction in the water uptake, which was significant to improve the durability of the epoxy.

In the Polymers, 2022, 14(10), 1935, Kim et al. investigated the effects of carbon fibers and graphite flakes on the composite materials’ heat dissipation properties and mechanical strength with various hybrid ratios in the matrix. When the ratio of carbon fiber and graphite powder was 30 phr and 20 phr, respectively, the highest thermal conductivity and thermal diffusivity were observed, and the lowest specific heat value was observed. It was also confirmed that thermal and mechanical properties significantly decreased as carbon fiber and graphite powder decreased and increased, respectively.

In the Composite Structures, 2022, 293, 115719, a wedge-extrusion bond anchorage system was applied to provide the reliable anchorage load-bearing capacity for CFRP plate. A new prestressed tension device was put forward to realize the coupling exposure of elevated temperature, distilled water and sustained loading. The mechanical and thermal properties of CFRP plate were conducted to obtain the long-term evolution exposed to the above environments. The results showed that the long-term life prediction showed the residual tensile strength retentions of CFRP plates were more than 50% for the service life of 30 years in civil engineering structures. The higher exposure temperature (60 C) gave rise to an additional degradation rate of ~17% compared to the room temperature (20 C).

In the revised manuscript, we cited the above significant works. Please see the References [5, 10, 11].

3) Is the non-covalent physical modifications reliable to improve the performance of composite? How to verify the long-term reliability for the physical modifications? Generally speaking, the chemical modifications through the formation of chemical bonds may be more stable and reliable.

√Response: We reviewed articles about the long-term reliability of composites with the help of non-covalent physical modifications, focusing on Triton™ X-100.

Meng et al. [R1] proposed a simple yet effective method to prepare multifunctional, durable and highly conductive graphene/sponge nanocomposites for multi-field application. By modifying graphene using the Triton™ X-100 and adjusting its fractions, they optimized the composite structure for high sensitivity, stable fatigue performance, less response time and ideal responses to temperature and pressure. George et al. [R2] also demonstrated the improved thermal reliability Paraffin/PANI composite modified by Triton™ X-100. Kaleemullah et al. [R3] investigated the effect of surfactant treatment on the thermal stability and mechanical properties of CNT/polybenzoxazine nanocomposites, and stated that the dimensional stability of the composites in terms of the coefficient of thermal expansion measured below Tg was much better in the CNT/polybenzoxazine nanocomposites than in the epoxy counterpart. Hrushikesh et al. [R4] studied the influence of covalent and non-covalent modification of graphene on the mechanical, thermal and electrical properties of epoxy nanocomposites. Covalent and non-covalent modification of graphene filled epoxy nanocomposites shows the significant improvement in mechanical, thermal and electrical properties compared to neat epoxy and graphene filled epoxy nanocomposites.

In summary, the non-covalent physical modifications represented by Triton™ X-100 were able to improve the performance of the composite with reliability. However, we have to admit the lack of detailed mechanisms studies before the improved results. Thanks for your forward-looking opinion.

[R1] Meng Q, Yu Y, Tian J, et al. Multifunctional, durable and highly conductive graphene/sponge nanocomposites[J]. Nanotechnology, 2020, 31(46): 465502.

https://doi.org/10.1088/1361-6528/ab9f73

[R2] George M, Pandey A K, Abd Rahim N, et al. A novel polyaniline (PANI)/paraffin wax nano composite phase change material: Superior transition heat storage capacity, thermal conductivity and thermal reliability[J]. Solar Energy, 2020, 204: 448-458.

https://doi.org/10.1016/j.solener.2020.04.087

[R3] Kaleemullah M, Khan S U, Kim J K. Effect of surfactant treatment on thermal stability and mechanical properties of CNT/polybenzoxazine nanocomposites[J]. Composites Science and Technology, 2012, 72(16): 1968-1976. https://doi.org/10.1016/j.compscitech.2012.08.020

[R4] Kulkarni H B, Tambe P, M. Joshi G. Influence of covalent and non-covalent modification of graphene on the mechanical, thermal and electrical properties of epoxy/graphene nanocomposites: a review[J]. Composite Interfaces, 2018, 25(5-7): 381-414.

https://doi.org/10.1080/09276440.2017.1361711

4) Please add the basic performance and parameters of raw materials and other relevant information in Part 2.1.

√Response: By looking up the handbook of production scheduling or testing these on our own, we supplemented a new Table 1 in 2.1. Materials section. Herein, we provided some important information for epoxy resin, hardener, GNPs and CNTs, which were related to the topic of the current work. On the other hand, we also added the producer and country belonging to the raw materials, experimental instrument, and software in the revised manuscript.

5) Figure 2 and table 1-2 shows the XPS curves and data before and after the modification. They should belong to the result part and should be put in the result and discussion part. This part focuses on sample preparation and performance test methods.

√Response: In the revised manuscript, we have adjusted Figure 2 and Tables 1-2 to section 3. Results and discussion as 3.1. Modification analysis along with a new FT-IR analysis.

6) How to determine the addition content of nanoparticles in Section 2.3? Is there any relevant reference basis?

√Response: Before this work, we investigated the electrical properties of several kinds of carbon nanofiller filled nanocomposites based on the percolation theory, such as [R5-R8]. We also characterized the dispersion state of different carbon nanofillers within the polymer matrix using the SEM instrument. We found that there existed a suitable nanofiller content range to ensure the improved effects for the nanocomposite. If the filler content was too low (percolation theory was determined to be ~0.1 wt%), the nanofillers were hard to form a continuous conductive network. In contrast, if the filler content was too high (above approximately 1.0 wt%), the nanofiller was easy to aggregate within the polymer matrix, which damaged the functional and mechanical properties of the as-prepared nanocomposites. In the current work, the main objective of this work is to endow the conductive functionality to nanocomposite specimen, as well as the improved mechanical properties to realize the overall designability of structure and function. Thus, we determined the GNPs content to be 0.1, 0.25, 0.50, and 1.0 wt% in 2.3. Specimen fabrication section. On the other hand, the total filler content of hybrid EP/CNT@X+GNPs nanocomposite was determined to be 0.5 wt% for comparison conveniently with EP/GNPs_0.5. The weight ratios of CNT@X to GNPs were varied from 3/7, 5/5 to 7/3 to understand their synergistic effects on the hybrid nanocomposite system. Based on the above discussion, we revised the 2.3. Specimen fabrication section to explain.

[R5] Wang F, Zhang K, Liang W, et al. Experimental and analytical studies on the flexible, low-voltage electrothermal film based on the multi-walled carbon nanotube/polymer nanocomposite[J]. Nanotechnology, 2018, 30(6): 065704. https://doi.org/10.1088/1361-6528/aaf195

[R6] Wang F, Tay T E, Sun Y, et al. Low-voltage and-surface energy SWCNT/poly (dimethylsiloxane)(PDMS) nanocomposite film: Surface wettability for passive anti-icing and surface-skin heating for active deicing[J]. Composites Science and Technology, 2019, 184: 107872. https://doi.org/10.1016/j.compscitech.2019.107872

[R7] Sun Y, Wang Y, Liang W*, F.X Wang* et al. In Situ Activation of Superhydrophobic Surfaces with Triple Icephobicity at Low Temperatures[J]. ACS Applied Materials & Interfaces, 2022, 14(43): 49352-49361. https://doi.org/10.1021/acsami.2c15075

[R8] Wang F X, Liang W Y, Wang Z Q, et al. Preparation and property investigation of multi-walled carbon nanotube (MWCNT)/epoxy composite films as high-performance electric heating (resistive heating) element[J]. Express Polymer Letters, 2018, 12(4).

https://doi.org/10.3144/expresspolymlett.2018.26

7) In addition to the macro performance testing, the micro performance testing and analysis are also critical for revealing the influence of nanoparticles on the properties of composites. It is suggested to add the test methods and details of the scanning electron microscopy.

√Response: In the revised manuscript, we divided a new 2.4. Characterizations and measurements section to describe the test method and details of not only SEM but also XPS, FT-IR, ultrasonic pulse-echo instrument, IR camera, and sourcemeter. Thanks for this professional comment.

8) Figure 5 shows that the load-displacement curves of composites through quasi-static 3P-ENB test. However, the control strength and corresponding improvement proportion should be quantified into a table, so that the reader can understand the relevant information more intuitively.

√Response: In the revised manuscript, we supplemented a new Table 4 to help the reader understand the relevant information more intuitively. Thanks!

9) Fig. 6 shows that the fracture toughness of the composite is improved to different degrees after adding different nanoparticles. Please provide some explanations of the improvement mechanism. It would be better if you could provide the mechanism diagrams.

√Response: In the revised manuscript, we supplemented a schematic diagram showing the interaction of the crack front with GNPs and CNT@X+GNPs particles, combined with the SEM images, as shown in Figures 8 and 9. The CNT@X+GNPs form a loosely 3-D structure to prevent the crack propagation across the adjacent GNPs.

10) In part 3.2.1 and 3.2.2, does the strain rate have a significant effect on the stress-strain curve of material? Why do you study the effect of strain rate?

√Response: With respect to Figures 10 and 12 of the revised manuscript, the nanocomposite specimens showed different stress-strain behaviors under the two strain rates of 500/s and 1000/s, studying the systematic effects of filler types and contents for the as-prepared nanocomposite based on a SHPB equipment. One can find that the compressive strength of every EP/GNPs specimen was reinforced to varying degrees due to the existence of GNPs particles, as shown in Figure 10a. Unfortunately, most of these reinforcement events were replaced by the sacrificial failure strain, indicating that the GNPs contributed more to enhancing the strength, not to toughness. With the addition of CNT@X into the nanocomposite system, the dynamic compressive properties of the EP/CNT@X+GNPs category increased obviously, including the compressive strength and failure strain (Figure 10b).

On the other hand, with an increasing strain rate of 1000/s, the linear stage of stress-strain curves under a higher strain rate became more obvious and adjacent, meanwhile the nonlinear stage decreased along with the higher failure strength, when compared to the stress-strain curves under a strain rate of 500/s (Figure 10). Noting that, different from the dynamic compressive behaviors under a strain rate of 500/s, although the compressive strength increased obviously, the failure strain decreased simultaneously for hybrid CNT@X+GNPs nanocomposite (Figure 13). The above discussion demonstrated that the 3-D CNT@X+GNPs filler constructed by 1-D CNT@X and 2-D GNPs within the epoxy matrix was suitably subject to the dynamic loading with a favourable strain rate effect.

11) The conclusions should be further cleared with some quantitative results based on the important phenomena and findings in this paper. 3~4 conclusions are suggested.

√Response: According to this comment, we rewrote the.4. Conclusions section in the revised manuscript and listed four important results with more quantitative data. This is a very helpful comment, thank you!

Thank you very much for careful and professional review! If you have any other questions, please let us know.

Round 2

Reviewer 2 Report

The authors considered most of the comments or adequately responded to the remarks contained in the review; therefore, the work may be approved for publication.

Reviewer 3 Report

Accepted!